# Radon and Lung Cancer: Current Trends and Future Perspectives

**DOI:** 10.3390/cancers14133142

**Published:** 2022-06-27

**Authors:** Mariona Riudavets, Marta Garcia de Herreros, Benjamin Besse, Laura Mezquita

**Affiliations:** 1Medical Oncology Department, Gustave Roussy Cancer Campus, University Paris-Saclay, F-94800 Villejuif, France; mariona.riudavets@gustaveroussy.fr; 2Medical Oncology Department Hospital Clínic i Provincial de Barcelona, IDIBAPS, 08036 Barcelona, Spain; garciadehe@clinic.cat (M.G.d.H.); lmezquita@clinic.cat (L.M.); 3Laboratory of Translational Genomics and Targeted Therapies in Solid Tumors, IDIBAPS, 08036 Barcelona, Spain; 4Department of Medicine, University of Barcelona, 08007 Barcelona, Spain

**Keywords:** lung cancer, radon, carcinogenesis, driver genomic alterations, non-smokers

## Abstract

**Simple Summary:**

Radon represents the main risk factor of lung cancer in non-smokers and the second one in smoking patients. In Europe, there are several radon-prone areas, but regulatory policies may vary between countries. Radon causes DNA damage and high genomic tumor instability, but its exact carcinogenesis mechanism in lung cancer remains unknown. Molecular drivers in NSCLC are more often described in non-smoker patients and a potential association between radon exposure and oncogenic-driven NSCLC has been postulated. This is an updated review on indoor radon exposure and its role in lung cancer carcinogenesis, especially focusing on its potential relation with NSCLC with driver genomic alterations. We want to contribute to rising knowledge and awareness on this still silent but preventable lung cancer risk factor.

**Abstract:**

Lung cancer is a public health problem and the first cause of cancer death worldwide. Radon is a radioactive gas that tends to accumulate inside homes, and it is the second lung cancer risk factor after smoking, and the first one in non-smokers. In Europe, there are several radon-prone areas, and although the 2013/59 EURATOM directive is aimed to regulate indoor radon exposition, regulating measures can vary between countries. Radon emits alpha-ionizing radiation that has been linked to a wide variety of cytotoxic and genotoxic effects; however, the link between lung cancer and radon from the genomic point of view remains poorly described. Driver molecular alterations have been recently identified in non-small lung cancer (NSCLC), such as somatic mutations (*EGFR*, *BRAF*, *HER2, MET*) or chromosomal rearrangements (*ALK*, *ROS1*, *RET*, *NTRK*), mainly in the non-smoking population, where no risk factor has been identified yet. An association between radon exposure and oncogenic NSCLC in non-smokers has been hypothesised. This paper provides a practical, concise and updated review on the implications of indoor radon in lung cancer carcinogenesis, and especially of its potential relation with NSCLC with driver genomic alterations.

## 1. Introduction

Lung cancer is the second most common type of tumor in the world. According to the Global Cancer Observatory, a project developed by the International Agency for Research on Cancer (IARC, part of the World Health Organization–WHO), during 2020 a total of 19,292,789 new cases of cancer were documented, led by breast cancer (11.7%) and followed by lung cancer (11.4%), which is the most common type of tumor in men (14.3%) [1].

In terms of mortality, lung cancer is the leading cause of death by cancer in the world, with 1,796,144 deaths registered in 2020 [1]. These data are a reflection of the serious health problem posed by this disease.

The main risk factor for developing lung cancer is tobacco smoking, which is related to 80–90% of cases, and responsible of most of the deaths [2,3,4].

In non-smokers, lung cancer comprises approximately 15–25% of all lung cancer cases and its epidemiology is currently less well developed. The IARC lists several environmental carcinogens, including indoor radon, air pollution, arsenic, chromium, asbestos, nickel, cadmium, beryllium, silica and diesel, among others [5]. Other described risk factors for developing lung cancer are chronic local inflammation [6,7] and a low consumption of fresh fruits and vegetables [8]. Family aggregation studies support the hypothesis of a multifactorial inheritance, although the mechanism of hereditary transmission has not been well described. However, recent evidence demonstrated that germline pathogenic variants in *EGFR* and *cMET* may play a role in lung cancer as well as in homologous recombination repair (HRR) genes such as *BRCA1*, *BRCA2* or *PALB2* [8,9,10].

Indoor radon was declared a human carcinogen in 1987 by the WHO and in 1988 by the United States Environmental Protection Agency (EPA). According to the WHO, radon may be responsible for 3–14% of lung cancer cases, which is considered the second leading cause of lung cancer in tobacco smokers and the leading cause in non-smokers [11]. In addition, radon accounts for around 21,000 deaths (2%) from cancer in Europe [11,12].

Here, we provide a complete, concise and updated review on radon gas and lung cancer focused on the biological and molecular perspective. The final objective of this review is to provide knowledge regarding lung cancer carcinogenesis, especially focusing on the potential relationship between radon and non-small cell lung cancer (NSCLC) harbouring driver oncogenic genomic alterations; and more importantly, to raise awareness of radon, as a preventable, but still silent risk factor of lung cancer.

## 2. What Is Radon?

Radon-222 (^222^Rn) is a radioactive gas that arises naturally as a decay product of uranium-238. Its indoor concentration is closely related with the uranium content of rocks on the earth’s crust beneath dwellings [13]. It has a half-life of 3.8 days, allowing it to diffuse through soil and into the air before decaying by emission of alfa-particles into a series of short-lived radioactive products, such as polonium-218 (^218^Po) and polonium-214 (^214^Po), and it also decays by emitting alfa-particles [13].

The main source of radon in air is soil, where radon concentrations are very high and reach 10,000 becquerels (Bq)/m^3^, especially in uranium ores, phosphate rock and metamorphic rocks such as granite, gneiss and schist [14]. On a global scale, it is estimated that 2.4 billion curies (90 EBq) of radon are released from the soil annually [15]. Radon releasing from the soil into the atmosphere depends on the Uranium-238 content of the geological substrate on which these buildings are settled, soil parameters (porosity, density and humidity) and weather conditions (wind, rain and humidity, etc.) [15].

Radon has also been identified in surface waters, where radon concentrations are less than 4000 Bq/m^3^. However, water from ground water systems can have relatively high levels of dissolved radon, however, and concentrations of 10,000,000 Bq/m^3^ have been reported in public water supplies [16].

Outdoor radon concentrations are relatively low and change daily, even in the same area, but can build up indoors. The highest concentrations to which workers have been routinely exposed occur underground, particularly in uranium mines [14]. Radon gas enters buildings through cracks, crevices and leaks that occur in foundations and connections between different materials in the building. The pressure inside buildings is usually lower than the pressure in the subsoil, making radon attracted inside by diffusion from the subsoil [11].

Indoor radon is the most important source of natural radiation (about 50%), to which humans are exposed [17].

As mentioned before, there is a huge variability in radon concentrations in the same geographic areas, and the available radon maps are based on exposure risk estimations. Indoor radon concentration also varies depending on the construction of buildings, ventilation habits, seasonal changes and daily weather variations. Since radon is about nine times denser than air, its concentration tends to accumulate in lower building stories, basements and ground floors. The most important radon transport mechanism is pressure driven airflow (i.e., advection) from the soil to the occupied space [11]. Because of radon fluctuations, estimating the annual average concentration of indoor radon requires reliable measurements of mean radon concentrations [11]. Different measuring methods exist, but the most recommended one by the WHO is the track-etched alpha detectors during at least a period of three months [14]. Addressing radon is important both in construction of new buildings (prevention) and in existing buildings (mitigation or remediation). Regarding mitigation, radon concentrations in existing buildings can usually be reduced at moderate cost, for example, by increasing underfloor ventilation, radon wells or soil pressurisation systems [11,14].

## 3. Radon Recommendations, Regulations and Policies

Assessment and reduction of indoor radon concentration is one of the 12 recommendations of the European Code Against Cancer. Concretely in the ninth point of this list, it figures “*Find out if you are exposed to radiation from naturally high radon levels in your home, take action to reduce high radon levels*”. In this context, the WHO recommends indoor radon concentrations under 100 Bq/m^3^ [11,18].

In Europe, there are wide differences between countries in terms of exposure to radon in dwellings. Countries with large amounts of granite or uranium-rich soils generally have very high levels of radon. There are several radon-prone areas, such as the Bohemian Massif, the north-west of Spain, the Massif Central, the Fennoscandian shield, the Vosges Mountains, the Central Alps, the North of Estonia and certain volcanic structures in central Italy [19]. Although the International Commission on Radiological Protection (ICRP) and The Council of the European Union have recommended reference radon levels for dwellings and workplaces, just a few countries have enforced cut-off levels, each of them setting different concrete radon limits. For example, Germany has a recommended reference level of 250 Bq/m^3^, while Switzerland and Sweden of 400 Bq/m^3^ and Spain 300 Bq/m^3^ [20].

In 2006, the Joint Research Centre of the European Commission launched a project to map radon at the European level, as part of a planned European Atlas of Natural Radiation. Currently, this map includes data from 29 countries, covering a fair part of Europe [21]. Interestingly, more than 30% of the surface area of the countries participating in the European Indoor Radon Map present a median concentration above 100 Bq/m^3^, and 4.2% is above 300 Bq/m^3^ (Figure 1) [21].

The revised European Directive from 2013 regarding basic safety standards, obliged the European Union member states to establish a national action plan regarding the exposure to radon and the European Commission for Atomic Energy (2013/59/EURATOM) established a directive of not exceeding 300 Bq/m^3^ in European homes [11,21].

In the United States of America (USA), the EPA recommends radon concentrations below 150 Bq/m^3^, in Australia the recommended limit is 200 Bq/m^3^ and in Canada 800 Bq/m^3^. In Asia, South Korea has fixed the limit to 148 Bq/m^3^, while in China it is 300 Bq/m^3^ in existing buildings and 100 Bq/m^3^ for new buildings. The Radon Council in Japan has still not established an indoor radon exposure reference level [21,22].

In order to reduce the disease burden associated with radon, it is important that national authorities use methods and tools to prevent radon exposure and to identify populations exposed to high indoor radon concentrations (both at home or at work), who are at risk for developing lung cancer and could benefit from lung cancer screening programmes. For example, the NCCN American guidelines allow smoking citizens to enter screening programmes if they prove to be exposed to indoor radon concentrations above 200 Bq/m^3^ [23]. On the other hand, prevention of radon exposure in new buildings can be implemented through appropriate provisions in the construction phase or installing a radon proof barrier at ground level [24].

## 4. Radon Epidemiological Evidence in Lung Cancer

### 4.1. Miners Population

In 1913, it was hypothesised that radon and radon progeny induced a high incidence of lung cancer among the silver and uranium miners of Germany [25,26]. This relationship was further established by many investigators in subsequent years in underground-miner studies, and were also confirmed in non-smoking miners [27,28,29]. The Wismut cohort is the worldwide largest uranium miner cohort with almost 58,700 cases, which found a direct linear association between radon exposure and lung cancer [30] Lubin et al. pooled data from 11 cohort studies of radon-exposed underground miners (two studies with non-smokers) and found a linear relationship for cumulative radon exposure, suggesting that expositions at lower levels, such as in homes, would carry a greater risk of lung cancer [29]. Provided all these data, in 1987 indoor radon was declared a human carcinogen by the WHO.

### 4.2. General Population

Different meta-analysis and pooling studies have demonstrated a dose-response relationship between radon exposure and lung cancer. These studies are very heterogeneous due to different sample size, number and type of detectors used, gender inclusion and smoking status.

The European pooling study by Darby et al. is the most representative pooled analysis from 13 European case-controlled studies and demonstrated a linear and statistical increase of 16% (range, 5–31%) of lung cancer risk per 100 Bq/m^3^ of indoor radon [12]. In an American pooling study carried out by Krewsky et al., the odds ratio (OR) for lung cancer-increased risk with residential radon concentration was 1.11 (95% confidence interval [CI], 1.00–1.28) [31]. In the north-west of Spain, Torres-Duran et al. reported an OR of 2.42 (95% CI, 1.45–4.06) for lung cancer in subjects exposed to concentrations above 200 Bq/m^3^ [32]; and Lorenzo-Gonzalez found an increased OR for lung cancer in 523 individuals with radon exposure ≥ 200 Bq/m^3^ compared with those exposed to ≤100 Bq/m^3^ [33]. In Sweden, Pershagen et al. observed risks of 1.3 (95% CI, 1.1–1.6) and 1.8 (95%CI, 1.1–2.9) for individuals exposed to 140–400 and more than 400 Bq/m^3^ [34]. In Germany, in a cohort of almost 4000 cases and controls, the lung cancer risk was 1.57 (95% CI, 1.08–2.27), 1.93 (95% CI, 1.19–3.13) and 1.93 (95%CI, 0.99–3.77) for concentrations of 50–80, 81–140 and more than 140 Bq/m^3^, respectively [35].

Gathering this data together, in 2003 Pavia et al. published the first meta-analysis on the association of indoor radon and lung cancer, comprising 17 case-control studies, demonstrating a 24% increased risk of lung cancer in patients exposed to more than 150 Bq/m^3^, with an OR of 1.24 (95% CI, 1.11–1.38) [36]. More recently, a metanalysis including English and Chinese studies demonstrated a non-linear dose-response association between environmental radon exposure and the risk of lung cancer, with an exponential association for high occupational cumulative radon doses. The dose-risk model estimated a risk ratio of 1.26 (95% CI, 1.21–1.30) for 100 working level months and 1.51 (95% CI, 1.38–1.65) for 200 working level months, respectively [37]. The results provided by these previous studies suggest that radon increases the risk of lung cancer without a threshold and can be carcinogenic at any level (even below international guideline recommendations) depending on the individual susceptibility, years of radon-exposure, childhood exposure and exposure to other carcinogens such as tobacco smoking, pollution or asbestos.

Most of these papers associated radon with two histological subtypes of lung cancer, the small cell lung cancer and the squamous cell lung cancer, because most of the population included in the miner pooled studies were smokers. Later on, when the mentioned studies demonstrated an increased lung cancer risk in non-smokers, adenocarcinoma histology was also associated to radon. Thus, available data concerning the specific histological features of radon-induced tumours are scarce. Mezquita et al. first report that well and moderated differentiated histological grades were more frequent in patients exposed to radon higher concentrations (>148 Bq/m^3^), papillary histological pattern being the most commonly found, particularly in cases exposed to >200 Bq/m^3^ [38]. Nonetheless, there is still little information about the clinical characteristics of the subjects that will develop lung cancer associated with radon.

### 4.3. Radon in Non-Smokers with Lung Cancer

While most studies on radon and lung cancer have included current and former smokers, some studies were conducted exclusively in non-smokers, describing an increased risk of lung cancer in this subset of patients exposed to high radon levels in their dwellings. A Swedish pooling study, including 745 non-smoking cases (436 cases and 1649 controls) found an excess risk of lung cancer attributable to radon exposure in the cohort of non-smokers, and an excess relative risk of 10% per 100 Bq/m^3^ average radon concentration was estimated [39]. In the north-west of Spain, Torres-Duran et al. studied a cohort of 521 non-smoking participants (192 cases and 329 controls), with an OR of 2.42 (95% CI, 1.45–4.06) for lung cancer in subjects exposed to concentrations above 200 Bq/m^3^ [32]. In this review, Lorenzo-Gonzalez gathered all the studies that studied radon-induced lung cancer and included non-smokers, and found a relative risk from 1.1 to 1.73, with no negative association in any of the included studies [33].

It is worth remarking that some more ancient studies have suggested an inverse relationship between radon exposure and lung cancer risk. These studies were against the linear-non-threshold (LNT) hypothesis-based, risk-assessment paradigm, and in favor with the hormetic relative risk (HRR) model, based on which low-level radon radioactive progeny is credited for the activated natural protection (ANP) against lung cancer, including smoking-related lung cancer.

An example is a case-control study of the lung cancer risk from residential radon exposure in Massachusetts performed by Thompson et al. in 2008, involving 200 cases and 397 controls, in which the results indicated that for radon levels up to and somewhat exceeding the EPA’s action level of 4 picocuries/L of air (approximately 150 Bq/m^3^), the lung cancer *odds ratio* was < 1, implicating negative values for excess relative risk (not permitted under the LNT model), which translated into a reduction in lung cancer risk [40]. Some other earlier studies by Cohen et al. reported similar findings [41,42].

Particularly, early mining studies of lung cancer risk from radon are subject to limitations arising mainly from uncertainties in estimates of radon exposure and are confounded by other exposures, such as smoking. A limitation in calculating the risk of lung cancer attributed to radon is that in articles reporting on epidemiological studies (residential and occupational) of lung cancer risk from radon exposure, the excess relative risk estimates for lung cancer varied from −0.13 to 0.73 per 100 Bq/m^3^ for exposure to radon gas, with a mean excess relative risk of 0.13 per 100 Bq/m^3^. Limitations in the evaluation of differences in risk across subgroups of the population include low precision due to small numbers of lung cancer cases among non-smokers, women and younger age groups. They are likely to underestimate excess relative risk estimates in studies of residential radon by 50 to 100 per cent. Moreover, since thoron and its decay products can be a significant component of the total exposure in some specific situations (workplaces or dwellings), it can be an additional source of error in radon studies that do not distinguish radon and thoron contributions to the total exposure [43].

Nevertheless, although controversy derives from these studies with no impact of radon in lung cancer, preclinical and clinical evidence is robust enough to consider radon as a carcinogen of the first group for lung cancer, being considered nowadays as the second risk factor of lung cancer, after tobacco smoke.

## 5. Radon and Lung Cancer Carcinogenesis

Indoor radon plays an important role in the genesis of lung cancer. Although a complete description of the nature of radon role in the development of lung cancer is limited, it is related to the emission of alpha particles with a high capacity to damage the epithelium of the respiratory tract (Figure 2) [44].

Because of its half-life, ^222^Rn itself engenders little radiation risk. However, ^218^Po and ^214^Po are solid, and tend to be deposited on the bronchial epithelium, thus exposing cells to alfa-irradiation [44].

The first comprehensive study of the toxic effects of radon exposure on human health was reported in a book “Health Risks of Radon and Other Internally Deposited Alpha-Emitters: BEIR IV” [17].

### 5.1. Genomic Effects of Alpha Radiation

Alpha radiation releases a large amount of energy in a very short linear track alpha (high-energy transfer capacity, HET), which is more biologically significant than either beta or gamma radiations and reacts much more readily with deoxyribonucleic acid (DNA), generating oxidative stress (reactive oxygen species, ROS) and hydroxyl radical attack through radiolysis, despite their reduced penetrating capability [44,45,46,47].

When alpha particles are inhaled, they can impact the respiratory epithelium, which is especially sensitive to radiation, and produce multiple cytotoxic and genotoxic effects that favor carcinogenesis [44]. This results in large-scale molecular changes that include: DNA double-strand breaks, single point mutations, deletions, substitutions and chromosomal rearrangements [44,48]. The consequences of this genomic instability is the modification of the cell cycle, dysregulation of cytokines and the increased production of proteins associated with cell-cycle regulation, apoptosis and carcinogenesis [44].

These effects can vary depending upon a number of different factors including dose, frequency of dose, cell type, cellular conditions (such as cell-cycle stage at exposure time) as well as intra and inter signaling between neighbour cells [44].

#### 5.1.1. Preclinical Evidence

In vitro data coming from bronchial epithelial cells exposed to high dose of alpha radiation (^238^Pu), found significant increased mutations in the *TP53* tumor suppression gene, as well as in the locus of the hypoxanthine-guanine phosphoribosyl transferase (HPRT), responsible for the transcription of an enzyme that plays a central role in the generation of purine nucleotides through the purine salvage pathway [44,49].

Radon-induced mutations have also been studied in lung tissue from rats exposed to high radon concentrations that developed lung tumours. Dano et al. developed a model of radon-induced rat lung tumours using comparative genomic hybridisation, and found frequent losses in chromosomes 4q12–21, 5q11–33, 15q and 19q, which are homologous to human chromosomes. These regions contain tumor suppressor genes and proto-oncogenes such as *MET*, *CDKN2A*, *MTS2*, *FHIT*, *RB1* and *MYC* [50]. The genetic similarities between rat and human lung cancer may suggest common underlying carcinogenic mechanisms in both species. Bastide et al. also studied radon-induced lung tumour molecular characteristics in the rat and found the dysregulation of the INK4a/CDK4/RB1 pathway, which is associated with cell cycle control [51].

#### 5.1.2. Clinical Evidence

In recent years, several studies have analyzed the correlation between genomic alterations and the risk of developing lung cancer in patients with high residential radon exposure. Some authors have suggested that the carcinogenic effect of radon may be potentiated by the accumulation of hereditary polymorphisms in *GSTM1/GSTT1*, mutations in tumours suppressor genes such as *TP53* and germline pathogenic alterations in HRR genes [52,53,54,55]. Lim et al. observed that radon-high exposed tumours presented a greater proportion of genes involved in DNA damage and repair, such as *ATR*, *ATRX*, *BARD1*, *RAD50* and *SMARCA4*, as well as in *TP53* [55].

##### Radon-Induced Mutations

As mentioned before, radon can induce a wide range of mutations, including point mutations, deletions, insertions and chromosomal arrangements that lead to cell cycle disruption, up or down regulation of cytokines and proteins associated with cell-cycle regulation and carcinogenesis. Much of the evidence obtained related to radon’s carcinogenicity comes from studies performed on cohorts of uranium miners, which either analyze peripheral lymphocytes or lung tumor tissue. Most of them have identified certain *TP53* mutations and polymorphisms that were also associated with tobacco smoke [56,57]. Deletions in HPRT have also been linked to lung cancer in miners, non-miners and in vitro models [58].

Additionally, Chen et al. reported mutant *KRAS* overexpression in bronchial epithelial cells chronically exposed to radon, which is related to let-7 downregulation and oxidative damage [59].

##### Impaired Chromosomal Arrangements

Disruptions to normal chromosomal arrangement represent a major contribution to cellular mutagenicity and have been considered markers of exposure to environmental stressors with increased cancer risk. Significant increases in chromosomal arrangements, as well as sister chromatid exchanges and micronuclei formation have been observed in miners exposed to high radon concentrations when compared to the control group [44].

##### Epigenetic Effects and Transcriptomic Changes

Besides genetic alterations, epigenetic factors also play an important role in radon carcinogenesis, including DNA methylation, modification of histones and microRNA dysregulation [60]. Different authors have described a dysregulation of concrete miRNA (with the upregulation of miR-16, miR-15, miR-23, mirR-19 as well as the downregulation of let-7, miR-194, miR-373, miR-124, mirR-146, miR-369 and miR-652) that alter DNA methylation, cell cycle, oxidative stress, inflammation, oncogene suppression and malignant transformation in patients with lung cancer exposed to radon [60,61,62,63,64].

Recently, studies in lung cancer patients exposed to high radon levels have tried to identify genome alterations by next generating sequencing (NGS). Iwamoto et al. pointed that the *EPAS1* mutation may be a biomarker for the development of lung adenocarcinoma and Ran Choi et al. explored lung tumours of non-smoker patients using NGS, and identified that *CHD4*, *TSC2* and *AR* mutations were more frequent in individuals exposed to high radon concentration (>100Bq/m^3^) [65,66].

Loiselle et al. analyzed the transcriptome of human lung epithelial cells exposed to radon in vitro and found that *AKR1CR* underwent the greatest expression changes [67].

Despite the recent evidence of molecular alterations, just a little part of the study analyses the exome/transcriptome of lung tumours of patients with high radon exposition, and, to date, no clear radon-induced mutation hotspot has been identified and, in some cases, the lack of knowledge with regards to exposures at low doses and the fact that many of the biomarker studies have limited sample sizes, may potentially explain why some of the results between laboratory investigations are inconsistent [44].

Identification of a specific genetic radon-related marker would provide significant assistance to the elucidation of radon-induced carcinogenesis and could act as both a useful biodosimeter and an identifier of risk at typical exposures. Further investigations into a consistent genetic radon-molecular signature are required [44].

### 5.2. Effects of Alpha Radiation on the Immune System

Ionizing radiation and the effects on the immune system were first observed in the atomic-bomb survivors and Chernobyl accident [68]. In this population exposed to high doses of radiation, bone marrow stem cells were severely damaged, which caused the depletion of innate immune cells, acquired immunodeficiencies, medullar aplasia and thymus dysfunction [68]. Nonetheless, in a population chronically exposed to natural radiation, low doses of radiation may also affect and reshape the immune system and its interaction with tumor cells [69].

Besides DNA and changes in the lung epithelium, ionizing radiation induced by radon could also affect the immune system in the tumor microenvironment. The overproduction of ROS in the lungs caused by persistent radon exposure may cause oxidative stress, leading to pulmonary inflammation [55].

Furthermore, radon could enhance tumor immunogenicity, by increasing genomic instability and cluster mutations in the tumor cells. In line with this, a recent work has revealed that tumor mutational burden (TMB) was higher in patients exposed to more than 48 Bq/m^3^ versus ≤48 Bq/m^3^, in a cohort of 41 non-smoker patients with lung adenocarcinoma [55]. Tumors with high TMB (TMB-H) tend to present more immunogenic neoantigens. Thus, the TMB was validated as a predictive biomarker of immunotherapy response [70].

Concerning immune cells, an accumulating amount of evidence based on epidemiological and pre-clinical studies indicate that low dose exposures of radiation might directly impact immune functions and reshape the immune system either in a pro-inflammatory or anti-inflammatory pathway, depending on various parameters such as dose, performance status, genetic background, environmental stressors, etc. This abandons the classical paradigm that radiation is purely immune-suppressive [55,71,72,73].

Chronic low dose rate irradiation of mice induced the stimulation of innate immunity by enhancing the cytotoxicity of pre-stimulated NK cells, myeloid cell differentiation and activation, suppression of pro-inflammatory responses and a shift towards a Th2-type T cell phenotype due in part to radiation-induced gene expression alterations in CD4+ T cells [72,73]. By these mechanisms, low dose radiation can stimulate T-cell activity in the tumor microenvironment, and this may also increase tumor immunogenicity.

Studies carried out with people exposed to different scenarios of low dose radiation (>100 mSv) described low prevalence of T-helper 1 (Th1) and a switch to Th2 response, changes in telomere length, cycle cell regulation and alterations in the expression of genes involved in the immune system related pathways [74].

Although radon may act as a source of low dose ionizing radiation, the specific changes in immune cells or immune signatures in the tumor microenvironment have not been studied and remain unknown.

## 6. Radon, Tobacco and Other Carcinogens

The association between radon exposure and lung cancer has been widely reported, and the combined effect of radon and tobacco smoke is thought to be synergic and higher than additive, rising 20 to 25 times higher than the risk of death by lung cancer in smokers exposed to radon beyond 200 Bq/m^3^ [11,75,76,77].

In 2001, Lagarde et al. suggested the existence of a synergistic effect between indoor radon exposure and tobacco smoke with a relative excess risk of 0.29 (95% CI, 0.03–1.24). Moreover, a recent study has described that smokers exposed to radon below 37 Bq/m^3^ had a risk of developing lung cancer of 20.16 (95%CI, 3.4–118.6) [39,76,78].

This may be explained by the changes in airway geometry caused by the increased production of mucus due to smoking, as well as the disruption of normal function of the lungs caused by physiological changes (disappearance of cilia, decrease in radon progeny redistribution rate, increase in breathing frequency, etc.) which led to the accumulation of radon progeny in the lung tissue and to a several-fold increase in the absorbed dose when compared with the healthy lung [79].

In addition, tobacco smoke and radon act as co-carcinogens in the early phases of the carcinogenic process. Both of them generate ROS that interact with DNA thought hydroxyl radical attack and radiolysis leading to bulky DNA adducts, with DNA repair pathway saturation and increased apoptosis [76,79]. They are both associated to mutations in *KRAS* and *TP53*. McDonald et al. found a 31% of *TP53* mutations and a 39% of *KRAS* mutations in 52 lung cancer cases from uranium miners inducing bulky DNA adducts, with a DNA repair pathway saturation and increased apoptosis [76,79,80,81].

Very little is known about other carcinogens that could also potentiate radon carcinogenesis, such as pollution, asbestos or arsenic. Asbestos and arsenic, two prominent non-tobacco carcinogens, also generate ROS and reactive nitrogen species (RNS), as radon does in the early carcinogenic process. This results in genetic alterations and epigenetic effects, such as changes in DNA methylation, changes in miRNA and histone modifications [82]. Pollution particles in the air facilitate radon progeny, since the micro-aerodynamic particulate matter of the atmosphere is used as a transmission vector for radon. Indeed, there are studies reporting a positive association between ambient fine particles, daily mortality and high radon concentrations [83].

## 7. Radon and Driver Genomic Alterations in Lung Tumours

As has been said, approximately 15–25% of lung cancer cases occur in non-smokers. Recent research suggests that lung cancer in this subgroup could be a different disease, since these patients present a higher survival and a different age of onset with a greater proportion of adenocarcinoma histology. Furthermore, evidence advocates that the molecular profile of lung cancer differs by carcinogen exposure history, and thus that lung cancer in non-smokers arises via a different biological pathway than lung cancer in smokers [23,84,85].

Especially in NSCLC with non-squamous histology, there exists different genomic alterations, the majority of mutations (such as in *EGFR*, *BRAF*, *HER2*, *MET* genes) or chromosomal rearrangements (*ALK*, *ROS1*, *RET*, *NTRK*) that constitute activating oncogenic mechanisms for cell proliferation and cancer development. These molecular aberrations are drivers for targeted therapies, such as tyrosine-kinase inhibitors or monoclonal antibodies, which have indisputably improved both the prognosis and the quality of life of lung cancer patients, and is now a standard of care in oncogene-driven NSCLC [23,84,85]. Molecular drivers in NSCLC are more often observed in non-smoker patients; however, no risk factor has been identified as being linked to oncogenic-addicted lung cancer.

The idea that cellular damage induced by indoor radon exposure could lead to lung cancer with molecular genomic alterations has been hypothesised. To date, five works have assessed this hypothesis, and demonstrates high median radon concentrations in NSCLC patients harbouring *ALK* rearrangements, as *EGFR* or *BRAF* mutations among others, although with a poor statistical power given the limited study sample sizes. These studies are summarised in Table 1.

First, Taga et al., assessed this relationship in a cohort of 70 patients with NSCLC living in Missouri (United States of America), with only 24 of them exposed to radon and carrying EGFR mutations. In this work, researchers observed a lack of association; however, it was performed in an area without elevated radon concentrations, with a median of radon exposure of 46.5 Bq/m^3^ (range, 37–57) [86]. Mezquita el al. studied this association in 48 patients from Spain (Madrid) with NSCLC harbouring EGFR/*BRAFV600E* mutations, or *ALK* rearrangements. Out of 15 smokers, active smoking at diagnosis (*n* = 5, 62.5%) was associated with a lower radon concentration (*p* = 0.026); however, any correlation was observed between indoor radon concentration and molecular alterations, with an overall median radon concentration of 104 Bq/m^3^ (range, 69–160) [38]. Nevertheless, the *EGFR* subgroup median concentration was 96 Bq/m^3^ (range, 42–915), which is 20 Bq/m^3^ below the median of the two other subgroups [38]. In addition, indoor radon concentrations above the WHO recommendation were most common in the *ALK* and *BRAF* groups, compared to the *EGFR* group [38]. Ruano-Raviña et al. published a retrospective study comparing *EGFR-*mutant or *ALK*-positive vs. negative NSCLC cases in smoking patients living in a radon-prone area (Galicia, Spain), with a median radon exposition of 182 Bq/m^3^ (range, 11–2350) [87]. Although there were no differences in residential radon level by subtype of mutations, they were two-fold higher in patients with *EGFR* exon 19 deletions compared with patients with *EGFR* exon 21 (L858R) single-point substitution mutations (216 vs. 118 Bq/m^3^; *p* = 0.057), and *ALK*-positive patients presented two-fold residential radon levels compared with *ALK*-negative cases (290 vs. 164 Bq/m^3^, respectively) [87].

In the Radon France study, an ecologic study, the prevalence of molecular alterations has been positively correlated with the indoor radon risk area based on the official French map (*Institut de Radioprotection et de Sûreté Nucléaire*, IRSN, France) in a cohort of 116,424 patients with NSCLC tested for *EGFR*, *BRAF*, *HER2*, *KRAS*, *ALK*, *ROS1* on the 28 French Platform led by INCa (French National Cancer Institute). Moreover, the prevalence of driver alterations, linked to a low tobacco consumption (*EGFR*, *BRAF*, *HER2* and *ROS1*), was higher in the French regions with high radon exposure [88].

Another more recent study, the BioRadon France study, found a higher prevalence of oncogenic alterations (*EGFR/ALK/BRAF/HER2/KRAS*) in 3994 patients born in high radon-exposed areas according to the IRSN map. Although no significant difference was observed after adjustment by age, gender or smoking status, probably due to a limited number of cases harbouring molecular alterations, patients with lung cancer and molecular alterations lived in areas with radon concentrations above the median exposure levels in France, and in high-risk areas there was a significantly higher rate of lung cancer in non-smokers. Therefore, cumulative exposure to residential radon should be taken into account in future radon studies [89].

## 8. New Molecular Epidemiology Studies for Radon and Lung Cancer

According to all previous studies and data suggested by the Radon France study, a preclinical and clinical investigation line has been developed in order to define the genomic profile of lung cancer associated to radon exposure.

The RadoNORM consortium (“Towards effective radiation protection based on improved scientific evidence and social considerations–focus on radon and NORM”), a consortium of 57 centres from 22 countries is aimed to manage the risk derived from radon and NORM exposition in order to assure effective radiation protection based on improved scientific evidence and social considerations, so as to implement the European radiation protection Basic Safety Standards.

Among the different RadoNORM projects, a section will be focused on the study of indoor radon exposition and lung cancer development in three different cohorts: rats, miners exposed to occupational indoor radon and patients exposed to indoor radon, with the objective to describe the clinical and biological lung cancer profiles in order to eventually develop for the first time a radon-associated lung cancer molecular signature.

In all three cohorts, an extensive and deep molecular characterisation will be performed to identify common molecular patterns associated to alpha radioactivity. First, in collaboration with the *Commissariat à l’énergie atomique et aux énergies alternatives* (CEA, France), the RADON-Rats is an observational retrospective study including samples from 52 rats with radon induced-lung cancer. At a clinical level and with the participation of the Federal Office for Radiation Protection (BfS) in Germany, the RADON-Miners will retrospectively correlate the genomic profile of 30 German uranium miners (Wismut) with lung adenocarcinoma and occupation radon exposition.

Finally, the BioRADON is a European observational prospective study aimed to analyse the molecular changes in cancer pathways of 993 patients with NSCLC exposed to residential radon. The primary endpoint will be the correlation between the molecular profile and indoor radon. Comprehensive genomic analyses will be performed to define a radon-associated signature in humans that can be compared to the signatures determined in the miners and previous cohorts. This will be the largest study characterising a radon-associated NSCLC molecular signature in humans. BioRADON is a downstream project of EORTC-1553 SPECTA, the European Organization for Research and Treatment of Cancer (EORTC) translational research platform, that has been active in Europe from April 2020.

## 9. Future Perspectives

Radon effects can vary widely between patients according to their different individual predisposition. In this vein, it has been recently described that patients with NSCLC and *EGFR* mutations could harbour other germinal pathogenic molecular alterations, such as *TP53* in the Li Fraumeni Syndrome or other genes involved in the DNA repairing machinery, such as *BRCA* germline mutations [9,90]. More studies will be needed to assess the individual susceptibility to radon exposition.

Worldwide, air pollution has been postulated to be the second cause of lung cancer after tobacco smoking, explaining approximately 14% of lung cancer cases, similarly to the 3–14% of radon-related cases depending on the area and country [91]. The evaluation of radon in combination with other carcinogens remains to be evaluated; thiswill be a RadoNORM task that will soon take place with the participation of other ISRN teams, in which we will be able to contribute with the wealth of data coming from the clinical and translational aforementioned studies.

The BEIR IV report, in its description of health effects of radon exposition, has included other potentially radon-associated malignancies; however, evidence is weak. RadoNORM will also study the possible correlation between radon, cerebral tumours and leukaemia in the paediatric population [77].

BioRADON will open an innovative area of translational research on radioactivity and cancer genomics. The development of novel and accessible molecular diagnostic platforms, such as NGS techniques that analyse simultaneously multiple molecular alterations, will help to better characterise a radon-associated molecular signature.

Importantly, BioRADON will raise consciousness of this preventable risk factor and it will serve as an instrument to promote radon policies and strategies on cancer prevention. If a radon related-phenotype can be defined, we will be able to identify the susceptible population that could enter early detection programs or follow-up/surveillance in public health services. Moreover, the profile associated to radon-exposed patients would help in assessing whether radon, such as other carcinogens, has an impact on the evolution of cancer, on the response to treatment with certain drugs or even on the prognosis of the disease, improving the management of cancer patients.

## 10. Conclusions

Lung cancer is a public health problem and the first cause of cancer death worldwide. In recent years, moreover, an increased number of lung cancer cases in non-smokers has been documented.

The ionising environmental radiation represents a major epidemiological concern world-wide, with radon representing the main risk factor of lung cancer in non-smokers and the second factor in smoking patients, with synergistic effects in the latter. Radon causes DNA damage and high genomic tumour instability, but its exact carcinogenesis mechanism and relationship with lung cancer remains unknown.

It is time to expand knowledge about radon: to study its mechanisms of cellular damage, its potential long-term health consequences and its undeniable relationship with lung cancer. We hope that the ongoing studies, such as BioRADON, will provide new data on the role of indoor radon exposure in the molecular signature of lung cancer, especially in non-smokers, as well as in the clinical and biological characteristics of lung cancer. These studies will strengthen scientific knowledge on lung cancer carcinogenesis, providing relevant information on how radon affects the evolution of lung cancer and if there is any impact on its prognosis. They will also contribute to promoting radon policies and strategies on cancer prevention.

## Figures and Tables

**Figure 1 cancers-14-03142-f001:**
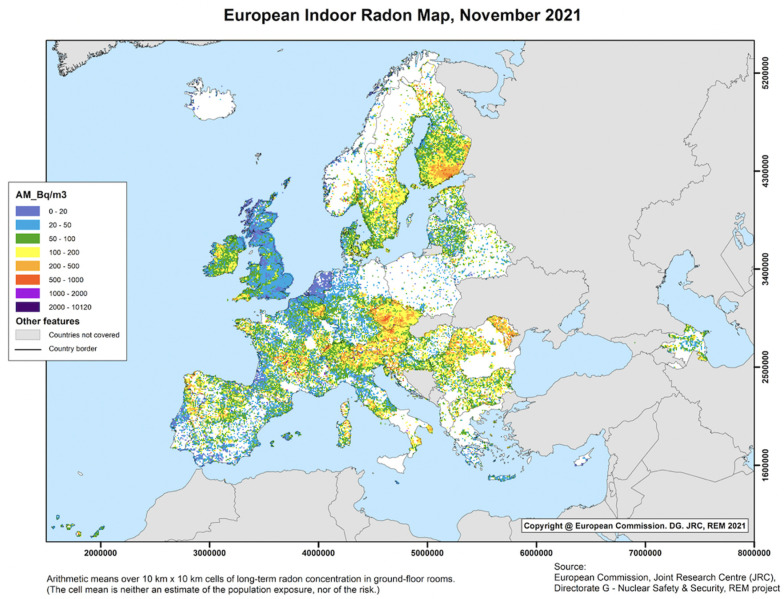
European Indoor Radon Map from the European Commission. Reprinted with permission from the European Atlas of Natural Radiation (EANR), https://remon.jrc.ec.europa.eu/About/Atlas-of-Natural-Radiation/Digital-Atlas/Indoor-radon-AM/Indoor-radon-concentration, accessed on 10 April 2022.

**Figure 2 cancers-14-03142-f002:**
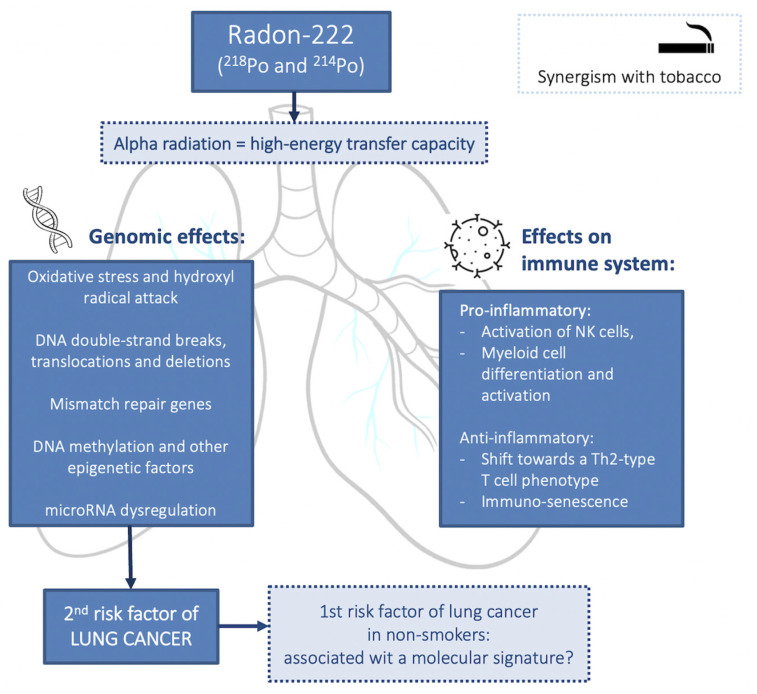
Radon-described mechanisms of carcinogenesis in lung cancer.

**Table 1 cancers-14-03142-t001:** Studies reported on the indoor radon concentration and molecular alterations in NSCLC patients.

Study	n Total of Patients	Type of Study	Molecular Alterations	Place of Radon Measurement	Studied Area	Radon Median Bq/m^3^ Exposition	Statistical Significance
Taga et al. [86] Prospective 2012	*n* = 70	Case control study	*EGFR*m ex19/21 (*n* = 24)	Current home	Non radon-prone area (Moussuri, USA)	*EGFR*m: 46.5 Bq/m^3^	Non sig. *p* = 0.16
Ruano-Raviña et al. [87] Retrospective 2016	*n* = 323	Case control study (subanalysis of previous study)	*EGFR*m * (*n* = 90) *ALK*r (*n* = 12)	Current home	Radon prone area (Galicia, Spain)	*EGFR*m ex19: 216 Bq/m^3^ *EGFR*m ex21: 118 Bq/m^3^ *ALK*r: 290 Bq/m^3^	Non sig. (*p* value non-available)
Mezquita et al. [88] Retrospective 2018	*n* = 116,424	Ecologic study **	*EGFR*m (*n* = 13,125) *ALK*r (*n* = 2928) *BRAF*m (*n* = 2419) *HER2*m (*n* = 816) *ROS1r* (*n* = 373) *KRAS*m (*n* = 27,314)	Current home	Non-radon prone, Intermediate and radon-prone area (France)	-	*p* < 0.0001
Mezquita et al. [38] Prospective 2019	*n* = 48	Cross-sectional study	*EGFR* m ^ (*n* = 36) *ALK*r (*n* = 10) *BRAF*m (*n* = 2)	Current home	Intermediate and radon-prone area (Madrid, Spain)	*EGFR*m: 96 Bq/m^3^ *ALK*r: 116 Bq/m^3^ *BRAF*m: 125 Bq/m^3^	Non sig. *p* = 0.238
Mezquita et al. [89] Retrospective 2021	*n* = 3994	Ecologic study **	*EGFR*m (*n* = 468) *ALK*r (*n* = 129) *BRAF*m (*n* = 89) *HER2*m (*n* = 32) *KRAS*m (*n* = 985)	Childhood home	Non-radon prone, intermediate and radon-prone area (France)	*EGFR*m: 72.49 Bq/m^3^ *ALK*r: 80.24 Bq/m^3^ *BRAF*m: 73.22 Bq/m^3^ *HER2*m: 72.74 Bq/m^3^ *KRAS*m: 71.79 Bq/m^3^	*p* = 0.0472

Abbreviations: *EGFR*m, *EGFR*-mutant; *BRAFm*, *BRAF*-mutant; *HER2*m *HER2-mutant; KRAS*m, *KRAS*-mutant; *ALKr*, *ALK*-rearranged; *ROS1*r, *ROS1*-rearranged; ex, exon; Non sig., non-significant; USA, United States of America.* Majority with *EGFR* exon 19 or 21 mutations; 5 cases were *EGFR* exon 20 mutations. ^ Including *EGFR* exon 18, 19 and 21 mutations. ** No radon measurement, estimation of radon exposure according to the National Map (IRSN).

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
