# Peer review of "Radon and Lung Cancer: Current Trends and Future Perspectives"

_cancers, 2022, doi:10.3390/cancers14133142_

Round 1
Reviewer 1 Report
The proposed manuscript is an interesting item in the discussion about the radon lung cancer risk. I generally recommend it for publication: it is clearly written and well understood. I think, however, that few remarks need to be considered prior to the publication:
1. My most important remark is connected with scientific objectivity. Authors stated that due to many studies and WHO/EPA recommendations, radon is a very strong carcinogen related to lung cancer. Especially the description in the last paragraph in Introduction section. Of course many studies, which confirmed such detrimental effects, were published, and many regulatory bodies (like WHO, IARC or EPA) followed them, but to keep the objectivity in the presented paper the Authors should discuss other findings, and take them into consideration, or at least inform that they do not agree with their conclusions (and why). This is also important from the public risk perception point of view. For example, some studies which showed no connection between low (!) concentration of radon and lung cancer:
- Thompson et al. Case-control study of lung cancer risk from residential radon exposure in Worcester County, Massachussetts. Health Phys. 2008
- Becker K. One century of radon therapy. Int. J. Low Radiation. 2004
- Pylak et al. Analysis of Indoor Radon Data Using Bayesian, Random Binning, and Maximum Entropy Methods. Dose-Response. 2021
- Scott BR. Residential radon appears to prevent lung cancer. Dose-Response. 2011
- Scott BR. Epidemiologic studies cannot reveal the true shape of the Dose-Response relationship for radon-induced lung cancer. Dose-Response. 2019
- Dobrzynski et al. Meta-analysis of thirty two case-control and two ecological radon studies of lung cancer. J Radiat Res. 2018
- Cohen BL. Test of the linear no-threshold theory of radiation carcinogenesis for inhaled radon decay products. Health Phys. 1995 [however, there is a strong critique of this study - please take care of that]
and so on. Therefore, in my opinion, the statement that radon is responsible for 2% cancer deaths in Europe, may be not true, because it is just a mathematical extrapolation and not every study confirmed that conclusion. More than that, the discussion about radon risk analysis cannot arbitrary exclude such independent studies and their opposite conclusions. Even UNSCEAR reports mentioned about that.
2. Paragraph no. 6, first sentence: it is difficult to compare immune system response to radon with atomic bombs because the latter is conneted with dose pulse (extremely high dose-rate) while radon is vely low dose-rate. Generally, dose-rate is of crucial importance in all risk analyses which is often omitted.
3. The same paragraph as previously, line no. 299-300: I recommend to add some deeper explanation about T-cell activity because it seems to be very interesting in this context.
4. Paragraph no. 10 (Future perspecives): you mentioned here about patients. However, the main focus of the manuscript is general public and public risk, not narrowed to lung cancer patients only. I feel some incoherency in wording here.
5. Line no. 456, simple typo: wea --> weak.
Author Response
REVIEWER 1
The proposed manuscript is an interesting item in the discussion about the radon lung cancer risk. I generally recommend it for publication: it is clearly written and well understood. I think, however, that few remarks need to be considered prior to the publication:
- My most important remark is connected with scientific objectivity. Authors stated that due to many studies and WHO/EPA recommendations, radon is a very strong carcinogen related to lung cancer. Especially the description in the last paragraph in Introduction section. Of course many studies, which confirmed such detrimental effects, were published, and many regulatory bodies (like WHO, IARC or EPA) followed them, but to keep the objectivity in the presented paper the Authors should discuss other findings, and take them into consideration, or at least inform that they do not agree with their conclusions (and why). This is also important from the public risk perception point of view. For example, some studies which showed no connection between low (!) concentration of radon and lung cancer:
- Thompson et al. Case-control study of lung cancer risk from residential radon exposure in Worcester County, Massachussetts. Health Phys. 2008
- Becker K. One century of radon therapy. Int. J. Low Radiation. 2004
- Pylak et al. Analysis of Indoor Radon Data Using Bayesian, Random Binning, and Maximum Entropy Methods. Dose-Response. 2021
- Scott BR. Residential radon appears to prevent lung cancer. Dose-Response. 2011
- Scott BR. Epidemiologic studies cannot reveal the true shape of the Dose-Response relationship for radon-induced lung cancer. Dose-Response. 2019
- Dobrzynski et al. Meta-analysis of thirty two case-control and two ecological radon studies of lung cancer. J Radiat Res. 2018
- Cohen BL. Test of the linear no-threshold theory of radiation carcinogenesis for inhaled radon decay products. Health Phys. 1995 [however, there is a strong critique of this study - please take care of that]
and so on. Therefore, in my opinion, the statement that radon is responsible for 2% cancer deaths in Europe, may be not true, because it is just a mathematical extrapolation and not every study confirmed that conclusion. More than that, the discussion about radon risk analysis cannot arbitrary exclude such independent studies and their opposite conclusions. Even UNSCEAR reports mentioned about that.
Answer:
We really appreciate this commentary from the reviewer. As said in the UNSCEAR report in 2019, both miner studies and residential studies of lung cancer risk from radon are subject to limitations arising mainly from uncertainties in estimates of radon exposure, particularly in early periods, and subject to confounding by other exposures, such as tobacco smoke. Limitations in evaluation of differences in risk across subgroups of the population include low precision due to small numbers of lung cancer cases among non-smokers, women and younger age groups. They are likely to underestimate excess relative risk estimates in studies of residential radon by 50 to 100 per cent. Moreover, since thoron and its decay products can be a significant component of the total exposure in some specific situations (workplaces or dwellings), it can be an additional source of error in radon studies that do not distinguish radon and thoron contributions to the total exposure.
We have clarified this point in the text by adding some new data.
Changes:
Page 6, Radon epidemiological evidence in lung cancer (Section 4), lines 229-259: It is worth to remark that some more ancient studies have suggested an inverse relationship between radon exposure and lung cancer risk. These studies were against the linear-non-threshold (LNT) hypothesis-based, risk-assessment paradigm, and in favor with the hormetic relative risk (HRR) model, based on which low-level radon radioactive progeny is credited for activated natural protection (ANP) against lung cancer including smoking-related lung cancer.
An example is a case-control study of lung cancer risk from residential radon exposure in Massachusetts performed by Thompson et al. in 2008, involving 200 cases and 397 controls, in which the results indicated that for radon levels up to and somewhat exceeding the EPA’s action level of 4 picocuries/L of air (approximately 150 Bq/m3), lung cancer odds ratio was <1, implicating negative values for excess relative risk (not permitted under the LNT model) translated into a reduction in lung cancer risk.39 Some other earlier studies by Cohen et al. reported similar findings.40,41
Particularly in early mining, studies of lung cancer risk from radon are subject to limitations arising mainly from uncertainties in estimates of radon exposure and subject to confounding by other exposures, such as smoking. A limitation in calculating the risk of lung cancer attributed to radon is that articles reporting on epidemiological studies (residential and occupational) of lung cancer risk from radon exposure, the excess relative risk estimates for lung cancer varied from −0.13 to 0.73 per 100 Bq/m3 for exposure to radon gas, with the mean excess relative risk of 0.13 per 100 Bq/m3. Limitations in evaluation of differences in risk across subgroups of the population include low precision due to small numbers of lung cancer cases among non-smokers, women and younger age groups. They are likely to underestimate excess relative risk estimates in studies of residential radon by 50 to 100 per cent. Moreover, since thoron and its decay products can be a significant component of the total exposure in some specific situations (workplaces or dwellings), it can be an additional source of error in radon studies that do not distinguish radon and thoron contributions to the total exposure42.
Nevertheless, although controversy derives from these studies with no impact of radon in lung cancer, preclinical and clinical evidence is robust enough to consider radon as a carcinogen of first group for lung cancer being considered nowadays as the second risk factor of lung cancer after tabaco smoke.
- Paragraph no. 6, first sentence: it is difficult to compare immune system response to radon with atomic bombs because the latter is connected with dose pulse (extremely high dose-rate) while radon is vely low dose-rate. Generally, dose-rate is of crucial importance in all risk analyses which is often omitted.
Answer:
Thank you for this remark. We agree that radiation coming from atomic bombs is extremely high and different from low dose radiation coming from radon. However, there are many studies describing immunological changes in the context of chronic exposure scenarios (A-bomb, Chernobyl, Kerala, etc.), in people who received low doses of radiation by living close to the affected area for a long period of time.
Lumniczky and colleagues gathered some of these studies from people who were exposed to chronic low dose ionizing radiation (in some studies <5 mGy) and report effects on the immune system. These effects can be either pro or anti-inflammatory, such as increase in Th2 response, changes in telomere length and dysregulation of inflammatory cytokines (Low dose ionizing radiation effects on the immune system. Environment International, 2021)
Text have been amended accordingly.
Changes:
Page 9, Effects of alpha radiation on the immune system (Section 5), lines 362-367: Concerning immune cells, an accumulating amount of evidence based on epidemiological and pre-clinical studies indicate that low dose exposures of radiation might directly impact immune functions and reshape the immune system either in a pro-inflammatory or anti-inflammatory pathway, depending on various parameters such as dose, performance status, genetic background, environmental stressors, etc. This abandons the classical paradigm that radiation is purely immune-suppressive [52,63–65].
- The same paragraph as previously, line no. 299-300: I recommend to add some deeper explanation about T-cell activity because it seems to be very interesting in this context.
Answer:
Thank you for this remark and appreciation. Following the reviewer suggestions, we have added some explanations in the text.
Changes:
- Page 9, Effects of alpha radiation on the immune system (Section 5), lines 359-367: Tumors with high TMB (TMB-H) tend to present more immunogenic neoantigens. Thus, the TMB has been validated as a predictive biomarker of immunotherapy response.
Concerning immune cells, an accumulating amount of evidence based on epidemiological and pre-clinical studies indicate that low dose exposures of radiation might directly impact immune functions and reshape the immune system either in a pro-inflammatory or anti-inflammatory pathway, depending on various parameters such as dose, performance status, genetic background, environmental stressors, etc. This abandons the classical paradigm that radiation is purely immune-suppressive [47,57–59].
- Page 9, Effects of alpha radiation on the immune system (Section 5), lines 374-377: Studies carried out with people exposed to different scenarios of low dose radiation (>100mSv) described low prevalence of T-helper 1 (Th1) and a switch to Th2 response, changes in telomere length, cycle cell regulation and alterations in the expression of genes involved in the immune-system related pathways [60].
- Paragraph no. 10 (Future perspecives): you mentioned here about patients. However, the main focus of the manuscript is general public and public risk, not narrowed to lung cancer patients only. I feel some incoherency in wording here.
Answer:
We thank the reviewer for this remark. Both patients with lung cancer and general public are targets and will benefit from ongoing studies aimed to determine a radon-associated molecular and clinical phenotype.
Some changes have been done to clarify this point.
Changes:
Page 13, Future Perspectives (Section 9), lines 538-549: BioRADON will open an innovative area of translational research on radioactivity and cancer genomics. The development of novel and accessible molecular diagnostic platforms, such as NGS techniques that analyze simultaneously multiple molecular alterations, will help to better characterize a radon-associated molecular signature.
Importantly, BioRADON will raise consciousness of this preventable risk factor and it will serve as an instrumental to promote radon policies and strategies on cancer prevention. If a radon related-phenotype can be defined, we will be able to identify the susceptible population that could enter early detection programs or follow-up/surveillance in public health services. Moreover, the profile associated to radon-exposed patients would help in assessing whether radon, like other carcinogens, has an impact on the evolution of cancer, on the response to treatment with certain drugs or even on the prognosis of the disease, improving the management of cancer patients.
- Line no. 456, simple typo: wea --> weak.
Corrections have been done accordingly in page 13, Future perspectives (Section 9), line 535.

Reviewer 2 Report
The authors wrote this review paper which aimed to discuss about Radon and lung cancer. This topic is broadly involved with environment, bio-molecular mechanism in cancer development and cancer prevention. The content of this manuscript is insufficient to be a review article published in the journal.
1. The risk factors of lung cancer for non-smoker are complex, and Radon is one of this risk factors. In the introduction section, authors should state clearly why they tend to discuss about Radon and lung cancer.
2. In the section 2 and 3, the introduction of Radon is too simple to let reader known what about Radon and its damage to human health.
3. The sections from 5 to 9 were not organized well, and the references cited were not updated. As a reader, I can not catch the point about the Radon and lung cancer development.
4. The future perspective and conclusion was not clear and concise, and they should let readers know about how to do cancer prevention and treatment regarding Radon.
Overall this manuscript is not qualified to be a review article for cancers.
Author Response
REVIEWER 2
The authors wrote this review paper which aimed to discuss about Radon and lung cancer. This topic is broadly involved with environment, bio-molecular mechanism in cancer development and cancer prevention. The content of this manuscript is insufficient to be a review article published in the journal.
- The risk factors of lung cancer for non-smoker are complex, and Radon is one of this risk factors. In the introduction section, authors should state clearly why they tend to discuss about Radon and lung cancer.
Answer:
We thank the author for this remark.
Radon is the second lung cancer factor risk and the first in non-smokers; therefore, promotion and education about indoor radon exposure regulations and its role in lung cancer should be one of the main objectives of the public health system.
This is a more focused review on how molecular epidemiology and the exhaustive characterization of the radon-associated lung cancer profile can impact cancer prevention; both from improving the level of knowledge in the general population as well as in our biomedical environment. For this reason, sections of generalities and epidemiology are simpler in order to further develop the part of the preclinical evidence and molecular data available today.
We have added some information accordingly.
Changes:
Page 2, Introduction (Section 1), lines 64-69: Here we provide a complete, concise and updated review on radon gas and lung cancer focused on the biological and molecular perspective. The final objective of this review is to provide knowledge regarding lung cancer carcinogenesis, specially focusing on the potential relationship between radon and non-small cell lung cancer (NSCLC) harbouring driver oncogenic genomic alterations; and more importantly, to raise awareness of radon, as a preventable, but still silent risk factor of lung cancer.
- In the section 2 and 3, the introduction of Radon is too simple to let reader known what about Radon and its damage to human health.
Answer:
Thank you for this remark. In these two sections, the idea was to introduce Radon-222, radon natural sources, outdoor and indoor radon, radon measurements and regulations. Radon risks to human health and carcinogenesis are discussed in sections 4 and 5. Following the reviewer suggestions, we have added some more information in section 2.
Changes:
Page 2, What is Radon? (Section 2), information has been added between line 77-111: The main source of radon in air is soil, where radon concentrations are very high and reach 10,000 becquerels (Bq)/m3, specially in uranium ores, phosphate rock and metamorphic rocks such as granite, gneiss, and schist [14]. On a global scale, it is estimated that 2.4 billion curies (90 EBq) of radon are released from soil annual [15]. Radon releases from soil into the atmosphere depends on the Uranium-238 content of the geological substrate on which these buildings are settled , soil parameters (porosity, density, humidity) and weather conditions (wind, rain, humidity, etc.) [15].
Radon has also been identified in surface waters, where radon concentrations are less than 4,000 Bq/m3. However, water from ground water systems can have relatively high levels of dissolved radon, however and concentrations of 10,000,000 Bq/m3 have been reported in public water supplies [16].
Outdoor radon concentrations are relatively low and change daily, even in the same area, but can build up indoors. The highest concentrations to which workers have been routinely exposed occur underground, particularly in uranium mines [14]. Radon gas enters buildings through cracks, crevices and leaks that occur in foundations and connections between different materials in the building. The pressure inside buildings is usually lower than the pressure in the subsoil, making radon attracted inside by diffusion from the subsoil [11].
Indoor radon is the most important source of natural radiation (about 50%) to which humans are exposed [17].
As mentioned before, there is a huge variability in radon concentrations in the same geographic areas and available radon maps are based on exposure risk estimations. Indoor radon concentration also varies depending on the construction of buildings, ventilation habits, seasonal changes, and daily weather variations. Since radon is about nine times denser than air, its concentration tends to accumulate in lower building stories, basements and ground floors. The most important radon transport mechanism is pressure driven airflow (i.e. advection) from the soil to the occupied space [11]. Because of radon fluctuations, estimating the annual average concentration of indoor radon requires reliable measurements of mean radon concentrations [11]. Different measuring methods exist, but the most recommended one by the WHO is the track-etched alpha detectors during at least a period of three months [14]. Addressing radon is important both in construction of new buildings (prevention) and in existing buildings (mitigation or remediation). Regarding mitigation, radon concentrations in existing buildings can usually be reduced at moderate cost, for example, by increasing underfloor ventilation, radon wells or soil pressurization systems [11,14].
- The sections from 5 to 9 were not organized well, and the references cited were not updated. As a reader, I can not catch the point about the Radon and lung cancer development.
Answer:
We appreciate this commentary and we do agree with the reviewer.
Some modifications have been done accordingly in the text. We have taken Figure 2 as reference to divide the different subsections in section 5 “Radon and Lung cancer carcinogenesis”.
We believe that “Radon, tobacco and other carcinogens”, “Radon and driver genomic alterations in lung tumours” and “New molecular epidemiology studies for radon and lung cancer” merit a special section each given its relevance in the present review.
Bibliography has also been updated.
Changes:
- Radon and lung cancer carcinogenesis
5.1. Genomic effects of alpha radiation
5.1.1. Preclinical evidence
5.1.2. Clinical evidence
5.1.2.1. Radon-induced mutations
5.1.2.2. Impaired chromosomal arrangements
5.2. Effects of alpha radiation on the immune system
- Radon, tobacco and other carcinogens
- Radon and driver genomic alterations in lung tumours
- New molecular epidemiology studies for radon and lung cancer
- The future perspective and conclusion was not clear and concise, and they should let readers know about how to do cancer prevention and treatment regarding Radon.
Answer:
We do agree with this reviewer’s comment. Some changes and additional information have been incorporated in order to make these sections clearer for the readers.
Changes:
- Pages 12-13, Future perspectives (Section 9), lines 525-549: Worldwide, air pollution has been postulated to be the second cause of lung cancer after tobacco smoking, explaining approximately 14% of lung cancer cases, similarly to the 3-14% of radon-related cases depending on the area and country. The evaluation of radon in combination with other carcinogens remains to be evaluated, what will be a RadoNORM task that will soon take place with the participation of other ISRN teams, in which we will be able to contribute with the wealth of data coming from the clinical and translational aforementioned studies.
The BEIR IV report, in its description of health effects of radon exposition, has included other potentially radon-associated malignancies, however evidence is weak. RadoNORM will also study the possible correlation between radon, cerebral tumours and leukemia in the pediatric population.61
BioRADON will open an innovative area of translational research on radioactivity and cancer genomics. The development of novel and accessible molecular diagnostic platforms, such as NGS techniques that analyze simultaneously multiple molecular alterations, will help to better characterize a radon-associated molecular signature.
Importantly, BioRADON will raise consciousness of this preventable risk factor and it will serve as an instrumental to promote radon policies and strategies on cancer prevention. If a radon phenotype can be defined, we will be able to identify the susceptible population that could enter early detection programs or follow-up/surveillance in public health services. Moreover, the profile associated to radon-exposed patients would help in assessing whether radon, like other carcinogens, has an impact on the evolution of cancer, on the response to treatment with certain drugs or even on the prognosis of the disease, improving the management of cancer patients.
- Page 13, Conclusion (Section 10), lines 564-567: These studies will strengthen scientific knowledge on lung cancer carcinogenesis, providing relevant information about how radon affects the evolution of lung cancer and if there is any impact on its prognosis. They will also contribute on promoting radon policies and strategies on cancer prevention.
Overall, this manuscript is not qualified to be a review article for cancers.

Reviewer 3 Report
The authors summarize the literature on radon and lung cancer very well, and it is well written as a review.
My concern is related to the phenomenon of cells becoming cancerous because of the damage to DNA caused by alpha rays from alpha decay. I say that alpha rays damaging DNA is a completely random phenomenon, and that the possibility of carcinogenesis caused by damaging other genes is more likely than the possibility of accidentally hitting EGFR, BRAF, or other mutations, which are the genetic mutations we now know about in lung cancer. I think it is more likely to be caused by damaging other genes. In other words, I think that radon does not have to correlate with known lung cancer gene mutations such as EGFR or ALK.
In any case, there is no clear answer at this point, and I think future research is awaited.
Round 2
Reviewer 2 Report
The authors revised the manuscript with some improvement in the quality, but I think they can do better with it. In section 5 "Radon and lung cancer carcinogenesis", I hoped the authors can expand the content and discuss more deeply in every small sections. Raise more updated references and summarized all the references briefly in their manuscript.
Author Response
REVIEWER 2
The authors revised the manuscript with some improvement in the quality, but I think they can do better with it. In section 5 "Radon and lung cancer carcinogenesis", I hoped the authors can expand the content and discuss more deeply in every small sections. Raise more updated references and summarized all the references briefly in their manuscript.
Answer:
We thank the author for this remark.
We have added some information as well as bibliography accordingly.
Changes:
- Page 7, Radon and lung cancer carcinogenesis (Section 5), lines 269-271: The first comprehensive study of the toxic effects of radon exposure on human health were reported in a book “Health Risks of Radon and Other Internally Deposited Alpha-Emitters: BEIR IV”[17].
- Page 7, Genomic effects of alpha radiation (Section 5.1), line 278: Alpha radiation releases a large amount of energy in a very short linear track alpha (high-energy transfer capacity, HET), being more biologically significant than either beta or gamma radiations, reacting much more readily with deoxyribonucleic acid (DNA), generating oxidative stress (reactive oxygen species, ROS) and hydroxyl radical attack through radiolysis, despite their reduced penetrating capability [44–47].
- Page 7, Genomic effects of alpha radiation (Section 5.1), lines 286-288: These effects can vary depending upon a number of different factors including dose, frequency of dose, cell type, cellular conditions (such as cell-cycle stage at exposure time) as well as intra and inter signaling between neighbour cells [44].
- Page 8, Radon-induced mutations (Section 5.1.2.1), lines 324-326: Additionally, Chen et al. reported mutant KRAS overexpression in bronchial epithelial cells chronically exposed to radon, related to let-7 downregulation and oxidative damage [59].
- Page 8, Impaired chromosomal arrangements (Section 5.1.2.2), lines 330-332: Significant increases in chromosomal arrangements, as well as sister chromatid exchanges and micronuclei formation have been observed in miners exposed to high radon concentrations when compared to the control group [44].
- Page 8, Epigenetic effects and transcriptomic changes (Section 5.1.2.3), lines 334-340: Besides genetic alterations, epigenetic factors play also an important role in radon carcinogenesis, including DNA methylation, modification of histones and microRNA dysregulation [60]. Different authors have described a dysregulation of concrete miRNA (with upregulation of miR-16, miR-15, miR-23, mirR-19 as well as downregulation of let-7, miR-194, miR-373, miR-124, mirR-146, miR-369 and miR-652) that alter DNA methylation, cell cycle, oxidative stress, inflammation, oncogene suppression, and malignant transformation in patients with lung cancer exposed to radon [60–64].
- Page 9, Genomic effects of alpha radiation (Section 5.1), lines 352-361: Despite the recent evidence of molecular alterations, just a little part of the studies analyses the exome/transcriptome of lung tumours of patients with high radon exposition, and, to date, no clear radon-induced mutation hotspot has been identified and, in some cases. The lack of knowledge with regards to exposures at low doses and the fact that many of the biomarker studies have limited sample sizes, may potentially explain why some of the results between laboratory investigations are inconsistent [44].
Identification of a specific genetic radon-related marker would provide significant assistance to the elucidation of radon-induced carcinogenesis and could act as both a useful biodosimeter and an identifier of risk at typical exposures. Further investigations into a consistent genetic radon-molecular signature are required [44].
Please see the attachment.

Round 3
Reviewer 2 Report
The authors had response to all my comments, and I don't have no more criticism.